# Dietary Factors Associated with Frailty in Old Adults: A Review of Nutritional Interventions to Prevent Frailty Development

**DOI:** 10.3390/nu11010102

**Published:** 2019-01-05

**Authors:** Juan José Hernández Morante, Carmelo Gómez Martínez, Juana María Morillas-Ruiz

**Affiliations:** 1Faculty of Nursing, Catholic University of Murcia, 30107 Murcia, Spain; csgomez@ucam.edu; 2LARES chair for social and health care of elderly people, LARES Nursing Home Association, 30500 Murcia, Spain; 3Food Technology & Nutrition department, Catholic University of Murcia, 30107 Murcia, Spain; jmmorillas@ucam.edu

**Keywords:** frailty, nursing home, protein supplementation, vitamin D, omega-3

## Abstract

Frailty syndrome is a medical condition that is characterised by a functional decline, usually from 65 years old on, and creates the need for assistance to perform daily living activities. As the population ages, the need for specialised geriatric care will increase immensely, and consequently, the need for specialised services for the care of these people will increase accordingly. From a nutritional point of view, to control or balance the nutritional status of residents will be essential in order to prevent sarcopenia and, consequently, frailty development. In this line, previous studies have highlighted the association among low energy intake, inadequate intake of protein and vitamin D, and an increased risk of frailty development. However, there is a lack of intervention studies on frail patients, especially in the realm of quality clinical trials. The few studies performed to date seem to indicate that there is a protective role of protein supplementation against frailty syndrome. In this regard, it is tempting to suggest daily 30 g protein supplements to prevent frailty. However, it is well established that excess protein can also be harmful; therefore, specific individual characteristics should be considered before prescribing these supplements. On the other hand, the relevance of other nutritional interventions, such as vitamin D, omega-3, and medium-chain triglycerides, is much more scarce in the literature. Therefore, we encourage the development of new clinical trials to carry out effective therapies to prevent frailty development.

## 1. Becoming Frail: An Upcoming Event

Although society and health systems are now facing health problems related to having food all the time, which causes issues such as obesity [1], little attention is being given to the next health food concern: Frailty syndrome, a medical condition characterised by a functional decline, usually from 65 years old, and that requires the need for assistance to perform the activities of daily living [2]. 

The prevalence of frailty is higher in older adults but is not considered part of normal ageing. One problem with frailty syndrome is how to differentiate normal from fragile ageing, that is, how to diagnose frailty. In this context, a key definition was provided by Linda Fried in 2001; a person suffers from frailty syndrome if three or more of the five following criteria are present: Weight loss, exhaustion, weakness, slowness, and inactivity [3]. Another problem with frailty syndrome derives from the term itself. Frequently, the term frailty is clinically used as either age-related comorbidity or disability, but these are not synonyms; in fact, comorbidity should be a step backward from frailty and dependence should be a step forward [4,5,6]. Finally, other trouble areas are that frailty develops in a continuum from previous stages and is characterised by clinically healthy subjects who have less strength to face external stressors before reaching the final stages in which older subjects have a very diminished functional capacity [7].

Therefore, considering the progressive ageing of the world’s population and the consequent inversion of the population pyramid [8], it is tempting to speculate that in a not too distant future, health providers will have to offer the greatest amount of resources for the care of geriatric patients, with all the consequences that this entails, such as providing more support to a greater number of people to carry out their basic daily activities [9,10].

In addition to its high prevalence, the clinical relevance of frailty syndrome lies in the increased morbidity, disability, and even mortality of these patients [11]. Evidently, sarcopenia, the loss of muscle mass, is a main factor involved in developing frailty [12]. Several authors refer to this syndrome as physical frailty [13]. Therefore, therapeutic approaches have focused on behavioural interventions to avoid the loss of muscle mass, such as nutritional and physical activity programmes; however, we will focus on the former interventions (Figure 1).

## 2. Role of Nursing Homes in the Nutritional Status of Residents

As the population ages, the need to generate more and more specialised services to care for these people will increase accordingly. Nursing homes represent the best example of these services [14]. Nursing homes are midway between social and health care, and they may cover all care spheres of geriatric care. On the one hand, these residences are a place to increase social relations and avoid social impairments, and on the other hand, they usually deliver health care [15,16].

From a nutritional point of view, nursing homes represent the perfect environment to control or balance the nutritional status of residents. Several reports have confirmed that the nutritional status of residents is not adequate but at least has significantly improved after entering a nursing home [17,18]. In fact, compared with community-dwelling elderly people, nursing home residents have a lower body mass index (BMI), present many nutritional deficiencies, and are predisposed to malnutrition [19]; thus, taking into account the nutritional control carried out in nursing homes, it is expected that the nutritional status should improve after admission. Furthermore, residents’ nutritional statuses used to be tightly monitored, allowing them to carry out the necessary interventions to balance their energy intake; therefore, malnutrition prevalence usually decreases after entering these residences [20,21]. 

Nursing home residents having the worst nutritional status is not unexpected. Community-dwelling people usually maintain adequate cognitive or functional capacity, but when these functions decrease, they usually move (or are moved) to nursing homes. The best way to avoid this impairment is to maintain a balanced nutritional status. Therefore, the beneficial effect of the nursing homes becomes greater in the long term [22,23].

However, as frailty syndrome increases, the need for specialised care at these institutions will also increase, which implies a specific nutritional intervention needed to prevent the onset and development of this syndrome. Considering this, next, we will focus on the dietary or nutritional interventions that have been carried out to date to prevent or treat frailty syndrome.

## 3. Dietary Factors Implied in Frailty Development 

How frailty develops is still a matter of intense debate, but there have been several factors noted, such as sarcopenia or muscle mass loss, which might be a cause. Here, nutritional status has been identified as a key factor in preventing the development of frailty syndrome. Several studies have found different associations between nutritional status, nutrient intake, and frailty development, but the factors that seem to exert a higher influence are calorie, protein, vitamin D, and calcium intake [24]. 

Among the studies focusing on the relationship between nutrition and frailty, the ‘Invecchiare (ageing) in Chianti’ (InCHIANTI) study deserves special attention [25]. In this community-based study, Bartali et al. described that low energy intake is associated with an increased risk (odds ratio (OR): 1.24) of frailty development. Moreover, they also described an increased risk of frailty when there is an inadequate intake of protein (OR: 1.98), vitamin D (OR: 2.35), vitamin E (OR: 2.06), vitamin C (OR: 2.15), and vitamin B9 (OR: 1.84). They also described that the sum of an inadequate intake of three nutrients was an independent and significant risk factor of frailty (OR: 2.12) [26]. 

A similar relationship between low protein intake and frailty was described elsewhere by several large cohort studies. In a study conducted on 24,417 older participants, Beasley et al. observed that those with the lowest risk of developing frailty were those with the highest protein intake [27]. Similarly, Tieland et al. showed that protein intake was significantly higher in community-dwelling than in frail elderly subjects, which may indicate an increased risk of frailty [28]. 

This high-protein preventive role is independent of the protein source, as described by Kobayashi et al. [29]. In their large cohort work, Kobayashi et al. showed that protein intake was inversely associated with frailty, regardless of the source of protein and the amino acid composition of the protein.

On the other hand, 25-hydroxyvitamin D (25OHD) levels have been proposed as an independent factor of frailty development. Boxer et al. first described this association in a small cohort study (*n* = 60) and concluded that 25OHD was an independent predictor of frailty [30]. Later, Chang et al. observed that frailty subjects with lower 25OHD levels were associated with higher odds of frailty (OR: 2.60) [31]. 

Finally, Ensrud et al., in a study conducted on 6307 women [32] and a subsequent study on 1606 men [33], reached a similar conclusion, although in these studies, the observed associations were much weaker than in the previous small cohort studies. In fact, in men, although low levels of 25OHD were independently associated with greater frailty prevalence at the baseline age, this did not predict a greater risk of frailty status at 4.6 years after the baseline age. 

In addition to these factors, several authors have described an inverse association between other nutrients, such as vitamin E, vitamin C, vitamin B6, and folate and frailty development [34,35,36]. 

Therefore, a balanced diet could be beneficial in avoiding frailty development. In this regard, adherence to a Mediterranean dietary pattern has been associated with lower odds of frailty. Data derived from a 6-year longitudinal study that was derived from a sub-cohort of the InCHIANTI study showed a lower risk (OR: 0.30) in those subjects with a high Mediterranean diet score [37]. Bollwein et al. described a similar relationship. Concretely, these authors showed that the subjects with the highest Mediterranean diet score were those with the lowest risk (OR:0.26) of frailty [38]. 

To reinforce these observations, in a previous paper that described the data of three longitudinal studies (the Seniors-ENRICA, three city Bourdeaux and integrated multidisciplinary approach cohorts), the authors observed a lower risk of frailty in subjects with higher fruit and vegetable consumption—taken in a dose–response manner at a daily consumption of three fruit and two vegetable servings [39]. In a recent systematic review, Lorenzo-Lopez et al. showed that the quality of the diet is inversely associated with the risk of being frail [40]. 

Before describing the nutritional interventions that can decrease the development of frailty syndrome, we would like to make a brief comment about the physical activity programmes conducted when studying frailty. Evidently, one way to prevent muscle loss is through physical exercise. Several clinical trials have evaluated the influence of a physical activity programme in frail older subjects, describing improvements in different areas, such as physical performance, depression, and cognition [41,42]; however, although fairly interesting, these studies are beyond the scope of the present review. We will focus solely on those trials dedicated to studying the effect of nutritional interventions. 

## 4. Dietary Interventions to Prevent Frailty

### 4.1. Mediterranean Diet

Since the Mediterranean diet was first described as a preventive factor against frailty, several authors have studied the association between various foods or components of this diet and the development of frailty. However, clinical trials to evaluate the relevance of this diet have not been conducted. 

However, Yarla et al. reviewed the relationship between olive oil intake, inflammation, and frailty. Interestingly, they suggested that the lower levels of inflammation-derived mediators, such as TNF-alpha and Il-6, produced by olive oil intake, can reduce the risk of frailty development [43]. This shows that although further studies are needed, the Mediterranean diet could be used for the prevention and treatment of frailty.

### 4.2. Specific Micronutrients

#### 4.2.1. Proteins

Although the ‘ideal’ diet for elderly people is not yet known, it seems clear that it is necessary to maintain muscle mass. Basically, there are two ways to do this: On the one hand, an individual can build new muscle mass, and on the other hand, an individual can stop the decrease of muscle. Evidently, the latter seems to be more interesting for the elderly population, specifically for frailty subjects. 

Although numerous clinical trials have been carried out to evaluate the effect of protein supplementation on muscle mass, in the present review, we focus solely on those trials performed to avoid the development or treatment of frailty syndrome, which are summarised in Table 1.

The first trial to this end was conducted in 2013 by Kim and Lee. These authors carried out a 12-week trial to evaluate the effect of a commercial liquid formula that added 400 kcal of energy, 25g of protein, 9.4g of essential amino acids, and 400 mL of water. In this study, an improvement in gait speed and “time-and-go” time was observed; however, grip strength was not significantly modified. The authors proposed the use of protein–energy supplementation to reduce frailty progression [44].

In 2016, Porter Starr et al. performed a similar work on obese older adults with low physical performance [45]. In this 6-month trial, subjects were divided into two groups: A group following a traditional hypocaloric treatment and one with higher protein intake (>30 g) at each meal. Interestingly, the authors showed that both groups lost a similar percentage of weight, and although there were no significant differences, the control group (hypocaloric diet) retained a greater amount of muscle mass. 

A similar study was also performed by Collins et al. [46]. In this 3-month study, which was also performed in 2016, subjects were assigned to whey protein and L-creatine co-supplementation or whey protein supplementation; however, this study did not have a control group without protein supplementation. Nevertheless, the authors described that both treatments were similar in improving physical activity performance.

In an interesting proposal for a clinical trial, Fernandes et al. described a study design composed of nine groups that could investigate different protein or amino acids supplementations with the additive effect of resistance training [47]. While waiting for the data from that trial, we consider it important to evaluate the summative effect of a nutritional intervention along with other interventions, such as resistance training. However, the physiological limitations of frailty may limit the effect of these interventions.

One of the few studies focused solely on protein supplementation in frail subjects was conducted by Niccoli et al. on hospitalised elderly populations [48]. In this hospital-stay-length trial, the authors randomised subjects into a control (no supplementation) group and whey protein supplementation group. Attending to the data obtained, the intervention group showed higher grip strength and knee extensor force, which indicates that the use of whey protein may be employed to improve protein nutritional status and rehabilitation outcomes in frail, elderly populations.

A study using protein supplementation to reduce frailty characteristics was also conducted by Dirks et al. In this small, randomised trial, frail subjects were divided into resistance-type exercise training supplemented with milk protein (15 g *bis in die*) or placebo. The outcome measure, muscle cross-sectional area, significantly improved compared with the placebo, and as a result, the intervention group patients were able to prolong their resistance-type exercise training [49].

To the best of our knowledge, the most recent study in the literature regarding protein supplementation and frailty prevention was a trial design performed by Vojciechowski et al. in 2018 [50]. This trial intended to divide subjects into five groups depending on the use of protein supplementation, physical performance and, very interestingly, the use of video game technologies, such as the Wii^®^ platform. 

While we await the results of these clinical trials, at the moment, there is very little evidence regarding the effect of protein supplementation on frailty, although the few studies seem to indicate a protective role of protein supplementation against frailty syndrome.

#### 4.2.2. Vitamin D

In addition to muscle mass preservation, many nutritional interventions in older adults are developed to keep their bone structures as healthy as possible. Several reports have demonstrated the relevance of dairy product intake to maintain a healthy bone status. Dehgdan et al. showed in a large multinational cohort study that a higher intake of dairy products was associated with lower death and cardiovascular disease risk [51]. 

Therefore, nutritional interventions have focused on both studies including the administration of dairy products or direct vitamin D and/or calcium administration. Latham et al. conducted a 10-week trial to evaluate the effect of vitamin D supplementation and high-intensity exercise compared with a control group. Unfortunately, the authors did not observe any effect of vitamin D on physical performance, which led them to not recommend this supplementation [52]. 

Boxer et al. conducted a 6-month clinical trial to evaluate the effect of vitamin D3 supplementation. A relatively small cohort (*n* = 64) was randomised and divided into weekly vitamin D3 50,000 IU supplementation or a placebo. However, as in the previous work, the authors described that vitamin D3 did not improve physical performance in their patients despite a robust increase in serum 25OHD. However, both groups received calcium supplementation, so it is not unreasonable to consider that this supplementation could be masking the effect of vitamin D [53].

More recently, Bauer et al. developed a study with a different methodology in which the effect of vitamin D and leucine-rich protein supplementation was studied. In this multicentre randomised work, community-dwelling older adults were divided into groups that would receive the supplementation or an isocaloric supplementation. Again, although a significant improvement was observed, there were no significant differences regarding the primary outcomes (handgrip strength and short physical performance battery scores) [54]. On the other hand, secondary outcomes, such as the chair–stand test and appendicular muscle mass, improved more in the intervention group. These observations led the authors to suggest that this intervention might benefit frail subjects, especially those who are unable to exercise [54]. 

#### 4.2.3. Omega-3

Other important nutrients that have been associated with muscle mass preservation are omega-3 fatty acids. Concretely, in older frailty subjects, León-Muñoz et al. described a positive association between higher omega-3 intake and a lower risk of frailty development. Through a 2-year large cohort longitudinal study, the authors described both the protective role of these fatty acids and the detrimental effect of a Westernised diet [55].

In line with this, Hutchins-Wiese et al. carried out a 6-month trial supplementing frail subjects with either a daily 2.4 g dose of eicosapentaenoic acid and docosahexaenoic acid or a placebo. The data revealed an improvement in walking speed in the fish oil group. However, the authors also suggested that a dietary intake of antioxidants (selenium and vitamin C) may interact with the fish oil to improve physical performance [56]. 

Another study conducted in 2016 by Strike et al. also employed omega-3 supplementation to reduce frailty symptoms. The participants received a daily multicomponent supplementation composed of docosahexaenoic acid, eicosapentaenoic acid, and other nutrients (phosphatidylserine, d-α tocopherol, folic acid, and vitamin B12) or a placebo. After a 6-month intervention, the authors showed that supplementation improved mobility in frail females, reinforcing the positive effect of these fatty acids; however, it cannot be ruled out that the other nutrients also played an important role in the observed effect [57].

#### 4.2.4. Other Interventions

Several reports have examined the combined effect of different supplementations. Ng et al. performed in 2015 a parallel-group, randomised controlled trial in which each participant received a nutritional supplement composed of iron and folate supplement, vitamin B6 and vitamin B12 supplement, and calcium and Vitamin D supplement taken daily for 24 weeks, which supposed a 20% increase in daily energy intake [58]. Attending to their observations, the frailty score was significantly reduced after intervention compared to the baseline score. It is important to highlight that the physical intervention also performed in this trial delivered a higher improvement in these patients, which reinforces the usefulness of combined therapies to prevent frailty [58].

The effect of a combined supplementation of L-leucine, cholecalciferol, and medium-chain (MCT) or long-chain triglycerides was analysed by Abe et al. in 2016. In this small cohort randomised trial, a significantly higher improvement of right-hand grip strength and walking speed was observed in the group receiving MCTs (6 g), leucine-rich amino acids, and the cholecalciferol group [59]. Therefore, the authors suggested that this family of triglycerides may be useful in preventing frailty development. Finally, the beneficial effect of L-carnitine in preventing frailty was studied by Badrasawi et al. [60]. In this 10-week trial, Badrasawi et al. observed that a 1.5 g/day L-carnitine supplementation could improve frailty scores and hand grip strength. Table 2 summarises the interventions other than protein supplements used to reduce frailty symptoms. 

## 5. Nondietary Interventions to Prevent Frailty

Other nutritional interventions in addition to dietary supplements should also be of interest to reduce frailty prevalence. In fact, as previously suggested [61], these interventions may be even more effective. 

In an interesting clinical trial, Wu et al. developed three interventions: Multiple micronutrient supplements, multiple micronutrients plus isolated soy protein supplements, and individualised nutrition education. The data showed that the only effective intervention was nutrition education, which reduced the frailty score by 60%, while the other interventions did not show any significant effect [61]. By contrast, the previous review of Manal et al. described no significant effect of nutritional education in old adults [63]. Probably, as the authors suggested, the baseline nutritional status of old adults (normal nutritional status or risk of malnutrition) may determine the effectiveness of this intervention.

A similar study was carried out by Chan et al. In a 3-month randomised clinical trial, community-dwelling older adults were divided into exercise and nutrition education or controls. From this study, the authors observed that exercise plus nutrition education can affect frailty status in the short-term, and in addition, they also described a significant long-term (12-month) improvement in bone mineral density [62]. Table 2 summarises the different nutritional interventions performed to avoid frailty development.

## 6. Conclusions

As the population ages, the frailty syndrome prevalence massively increases. This syndrome is characterised by a functional (physical and cognitive) decline, usually in adulthood, and requires a specific geriatric care. To date, there is no curative treatment for frailty, so the efforts have focused on the prevention and palliation of symptoms. In this regard, the interventions that have been described as effective are physical activity and nutritional interventions. However, it is discouraging to note the lack of quality clinical trials that have been conducted regarding frailty. In fact, although numerous nutrients have been tested, such as vitamin D, omega-3, medium chain triglycerides, etc., its effectiveness to date seems very limited. Additionally, to our knowledge, protein supplementation is the unique and well-defined intervention that may be able to prevent and treat frailty symptoms. In this regard, considering the previous reports conducted to date, it is tempting to suggest a daily 30 g protein supplements in old adults to prevent frailty. However, it is well established that excess protein can also be harmful; therefore, specific individual characteristics, like kidney or hepatic function, should be considered before prescribing these supplements. Nevertheless, we encourage the development of new clinical trials to carry out effective therapies to prevent frailty development.

## Figures and Tables

**Figure 1 nutrients-11-00102-f001:**
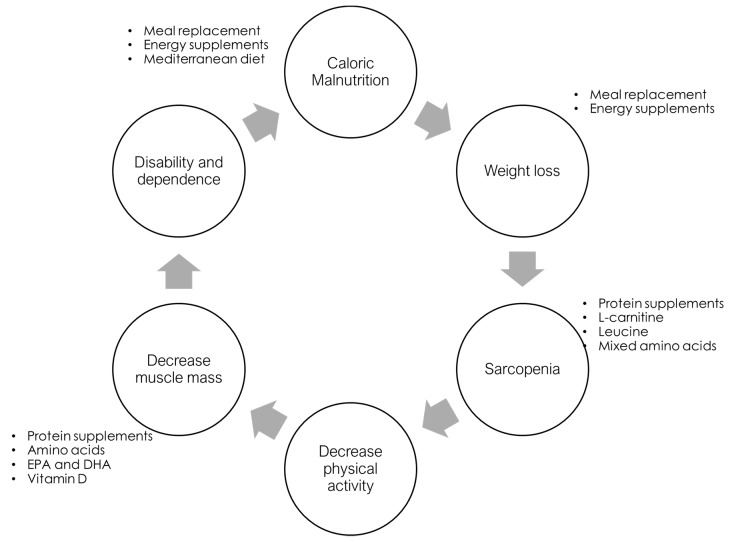
Frailty development cycle, adapted from Fried et al. [3] and possible nutritional targets to prevent frailty development. EPA: Eicosapentaenoic acid, DHA: Docosahexaenoic acid.

**Table 1 nutrients-11-00102-t001:** Summary of clinical trials performed to evaluate the effect of protein supplementation on frailty syndrome.

Reference	Population and Study Design	Age Range *	Intervention	Outcome Measure	Results
Kim et al. 2013 [44]	*n* = 87; Randomised controlled trial.	79 ± 6	Two 200-mL liquid formula (400 kcal, 25 g of protein, 9.4 g of essential amino acids, 400 mL of water) per day for 12 weeks	Change of the physical functioning and SPPB	Physical functioning increased by 5.9% in the intervention group
Porter Starr et al. 2016 [45]	*n* = 67; 6-month randomised controlled trial	68 ± 5	Regimen with higher protein intake (>30 g) at each meal	Physical function and lean mass	The increase in the protein content was greater than in the control (*p* = 0.02)
Collins et al. 2016 [46]	*n* = 18; a 14-week, double-blind, randomised, parallel-group, placebo controlled exploratory trial.	70 ± 5	Whey protein and creatine co-supplementation or whey protein supplementation	Muscle function and body composition	Both groups were similarly effective in improving muscle function
Fernandes et al. 2017 [47]	*n* = 90 (projected); a double-blind, randomised, placebo-controlled, parallel-group clinical trial	ND	Isolated leucine supplementation (study 1); protein source (whey vs. soy–study 2); combination of whey protein and creatine (study 3)	Muscle cross-sectional area, fibre cross-sectional area, body composition	Not finished yet
Niccoli et al. 2017 [48]	*n* = 47; Randomised clinical trial	82 ± 2	An oral dietary product containing 24 g of whey protein per day in addition to their usual diet	Frailty criteria	Whey protein significantly increases grip strength
Dirks et al. 2017 [49]	*n* = 34; Randomised, double-blind, placebo-controlled trial with 2 arms in parallel.	77 ± 1	6-month progressive resistance-type exercise training supplemented with milk protein (2 × 15 g/day)	Type I and type II muscle fibre specific cross-sectional area	Protein supplementation augmented muscle fibre hypertrophy following prolonged resistance-type exercise training in frail older people
Vojciechowski et al. 2018 [50]	A randomised controlled clinical trial with a sample of pre-frail older women (*n* not defined)	ND	Physical training combined with protein supplementation	Strength and power of the lower limbs and body composition	Not finished yet

* Age was described as mean age ± sd. ND: Not determined. SPPB: Short-form health survey.

**Table 2 nutrients-11-00102-t002:** Summary of the other clinical trials performed to evaluate the effect of nutrient supplementation or dietary interventions on frailty syndrome (protein supplement trials are summarised in Table 1).

Reference	Population and Study Design	Age Range *	Intervention	Outcome Measure	Results
Latham et al. 2003 [52]	*n* = 243; Multicenter, randomized, controlled trial with a factorial design	79 ± 2	Single dose of vitamin D (calciferol, 300,000 IU)	Physical health according to the short-form health survey (SPPB)	There was no effect of either intervention on physical health or falls
Boxer et al. 2013 [53]	*n* = 64; Parallel-design, double-blind randomised controlled trial	66 ± 10	Weekly Vitamin D3 50,000 IU	The primary outcome was peak oxygen uptake	Vitamin D3 did not improve physical performance
Bauer et al. 2015 [54]	n = 380; Multicenter, randomized, controlled, double-blind, 2 parallel-group trial	77 ± 7	A vitamin D and leucine-enriched whey protein nutritional supplement	Handgrip strength and SPPB score	Improvements in muscle mass among sarcopenic older adults
Hutchins-Wiese et al. 2013 [56]	*n* = 126; Randomized, double blind pilot study.	75 ± 6	2 fish oil (1.2 g eicosapentaenoic acid (EPA) and docosahexaenoic acid (DHA)) or 2 placebo (olive oil) capsules	Frailty assessment	Physical performance was significantly improved by fish oil supplementation
Strike et al. 2016 [57]	*n* = 27; Stratified block randomisation design	ND	1g DHA, 160 mg eicosapentaenoic acid, 240 mg Ginkgo biloba, 60 mg phosphatidylserine, 20 mg d-α tocopherol, 1mg folic acid, and 20 µg vitamin B12	Mobility assessed motion capture camera system	Multinutrient supplementation improved mobility in older females
Ng et al. 2015 [58]	*n* = 250; Randomised 5-arms clinical trial	70 ± 5	Combined nutritional supplement (iron, folate, vitamin B6, vitamin B12, vitamin D and calcium	Frailty status	Frailty index score was significantly improved in subjects supplemented with the combined nutritional supplement
Badrasawi et al. 2016 [60]	*n* = 50; Randomised, double-blind, placebo-controlled clinical trial	68 ± 6	L-carnitine	Frailty status	Frailty index score was significantly improved in subjects supplemented with L-carnitine
Wu et al. 2018 [61]	*n* = 40; 3-month, single-blind, parallel group, randomised controlled trial	73 ± 2	Multiple micronutrient supplements, multiple micronutrients plus isolated soy protein supplements and individualised nutrition education	Frailty score	Only individualised nutrition education decreases frailty score
Chan et al. 2012 [62]	*n* = 117; 3-month single site randomised controlled trial	71 ± 4	Exercise and nutrition (EN) or problem-solving therapy	Cardiovascular health study phenotypic classification of frailty	EN intervention resulted in short-term frailty status improvement

* Age was described as mean age ± sd. ND: Not determined.

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
