# Peer review of "Dietary Factors Associated with Frailty in Old Adults: A Review of Nutritional Interventions to Prevent Frailty Development"

_nutrients, 2019, doi:10.3390/nu11010102_

Reviewer 1 Report

This manuscript is an interesting look at nutritional interventions that may help combat the onset of frailty in the elderly population. However, the manuscript requires some modifications before publication:

1. The title "Dietary Interventions to Prevent Frailty" is a little misleading. Mainly because the section after talks about the association of frailty with eating patterns - not specific interventions. It is suggested that this part is reorganized to have a section based on dietary patterns and associations with frailty, and the associated discussion, before advancing to interventions. Make the distinction between the two much clearer.

2. Protein interventions are evaluated alongside amino acid interventions. However, single or mixed amino acid supplementation is distinctly different from protein supplementation. Amino acids should be considered separately from protein. Indeed, carnitine - an amino acid that is not found in proteins - should not be included in this table at all. This would need to be in a separate discussion.

3. Similarly, the table on vitamin D and omega 3 fatty acids should be split into the two interventions - it does not make sense to combine them.

4. The Kim and Lee paper on page 8 seems completely out of place - does this belong in the protein intervention section?

5. A graphical representation of the topic would be extremely helpful possibly to orient readers to the interventions being discussed and why.

Author Response

We would like to thank your comments. Your initial questions have been highlighted in bold, to differentiate from our comments described right below in the attached document.

Reviewer 2 Report

The time of writing narrative reviews is passed. Now we need systematic reviews that use sensitive and specific search methods... Those that cover both European, Asian, and American search engines. For example, Scopus, and PubMed should be both covered, We should also be given the terms and dates and the other aspects of the search strategy.... So, we need much more methods information before we are able to rely on the results of a review paper.  Some statistical strategies should also summarize the results of the review. Effect size of each study could appear in a graph, similar to the meta analyses. Please resubmit after revision.

Author Response

(The authors gave the same response as above.)

Reviewer 3 Report

This review article by Morante et al provides a comprehensive overview on nutritional interventions to prevent frailty in nursing homes. The topic is of significant interest to the readers in the field since there are relatively fewer studies that adequately address what interventions can be taken to combat frailty.

Overall, the review has been nicely written and structures. I have only minor comments:

1. Please cite the following articles:

(a) Nutritional determinants of frailty in older adults: A systematic review, López et al, 2017

(b) Nutritional, Physical, Cognitive, and Combination Interventions and Frailty Reversal Among Older Adults: A Randomized Controlled Trial, Tze Pin Ng et al, 2015

(c) Nutrition and Frailty: A Review of Clinical Intervention Studies, Manal et al, 2015

2. If information on the age range of the study subjects from the clinical trials cited are available, it would b helpful to include that in the table.

Author Response

(The authors gave the same response as above.)

Reviewer 4 Report

This is an excellent review of potential nutritional prevention methods for frailty syndrome. Currently proposed preventive dietary methods with clinical trial data were discussed. Clinical studies involving several nutrients and Mediterranean diet were discussed. I would agree that certain amount of protein supplementation seemingly to be one nutritional method to possibly reduce frailty syndrome rate, with other choices needing more research.

One small concern is the title and the abstract both provide an impression of this manuscript primarily reviewing on the prevention of frailty syndrome specifically in nursing home environment. However, most of the reviewed literature were not specifically carried in nursing homes. Then maybe the nutrition application in nursing homes should be a suggestion of this review instead of an conclusion. Actually, it seems protein supplement in elderly diet is the main focus of the conclusion for this review.

Also, a proposed amount of protein supplement in diet and potential adverse effects or concerns should be discussed as well. At least more comparison between the different protein diet among cited clinical trials can be added.

Overall speaking, excellent job!

Author Response

We would like to thank your comments. Your initial questions have been highlighted in bold, to differentiate from our comments described right below in the attached document.

Round  2

Reviewer 2 Report

The revision is satisfactory. The authors have explained why they could not do a systemic review.

Author Response

Thanks very much.